# On Computing Pairwise Statistics with Local Differential Privacy

**Badih Ghazi**
Google Research
Mountain View, CA, US
badihghazi@gmail.com

**Pritish Kamath**
Google Research
Mountain View, CA, US
pritish@alum.mit.edu

**Ravi Kumar**
Google Research
Mountain View, CA, US
ravi.k53@gmail.com

**Pasin Manurangsi**
Google Research
Bangkok, Thailand
pasin@google.com

**Adam Sealfon**
Google Research
New York, NY, US
adamsealfon@google.com

## Abstract

We study the problem of computing pairwise statistics, i.e., ones of the form $\binom{n}{2}^{-1} \sum_{i \neq j} f(x_i, x_j)$, where $x_i$ denotes the input to the $i$th user, with differential privacy (DP) in the local model. This formulation captures important metrics such as Kendall's $\tau$ coefficient, Area Under Curve, Gini's mean difference, Gini's entropy, etc. We give several novel and generic algorithms for the problem, leveraging techniques from DP algorithms for linear queries.

## 1 Introduction

Differential privacy (DP) [DMNS06] is a widely studied and used notion for quantifying the privacy protections of algorithms in data analytics and machine learning. It has witnessed many practical deployments [EPK14, Sha14, Gre16, App17, DKY17, Abo18, RE19, TM20, KT18, RSP+21]. On a high level, DP dictates that the output of a (randomized) algorithm remains statistically indistinguishable if the input of any single user is modified; the degree of statistical indistinguishability is quantified by the privacy parameter $\varepsilon > 0$.

An important setting of DP is the *local* model [EGS03, KLN+11], where each of $n$ users holds an input that they wish to keep private. An analyst wishes to compute some known function of the users inputs. The analyst and the users engage in an interactive protocol at the end of which the analyst is supposed to compute an estimate to the value of the function on the user inputs. When the aforementioned statistical indistinguishability property is enforced on the algorithm's transcript, the algorithm is said to be $\varepsilon$-local DP (see Section 2 for a formal definition). *Non-interactive* algorithms refer to the setting where each user sends a single (DP) message to the analyst, who is then supposed to output an estimate of the desired value without any further interaction with the users. As usual in the interactive local DP setting, we assume a broadcast model where every communication is visible to all users and to the analyst (and subject to the DP constraint). The number of rounds of an interactive protocol refers to the number of back-and-forths between the set of users and the analyst.

While the local DP setting offers a compelling trust model compared to the central DP model (where an analyst is assumed to have access to the raw user data and the privacy guarantee is only enforced at its output), the local setting is often limited by lower bounds on the error incurred by private protocols (e.g., [BNO08, CSS12]). In this work, we study some basic tasks in analytics and learning, and give local DP protocols with significantly smaller error than what was previously known.

37th Conference on Neural Information Processing Systems (NeurIPS 2023).

**Quadratic Form Computation.** Throughout the paper, we consider the following task:

**Definition 1** (Quadratic Form Computation). *Given a matrix $W \in \mathbb{R}^{k \times k}$. Each user $i$ receives $x_i \in [k]$. The goal is to compute the quadratic form $h_{\mathbf{x}}^T W h_{\mathbf{x}}$ where $h_{\mathbf{x}} \in \mathbb{Z}_{\geq 0}^k$ denotes the normalized histogram of the input, i.e., $(h_{\mathbf{x}})_b := \frac{1}{n} \cdot |\{i \in [n] \mid x_i = b\}|$.*

An algorithm computing quadratic forms immediately implies an algorithm for computing pairwise statistics (also known as $U$-*statistics of degree* 2) defined as $\mathcal{F}(X) = \frac{1}{\binom{n}{2}} \sum_{1 \leq i < j \leq n} f(x_i, x_j)$, where $f : \mathcal{X}^2 \to \mathbb{R}$ is a symmetric function (called *kernel*), and $x_i$ is the input to the $i$th user. The family of pairwise statistics is notable in that it contains several statistical quantities that are widely used in different areas, including the Area Under the Curve (AUC), Kendall's $\tau$ coefficient, the Gini's mean difference, and the Gini's diversity index (aka Gini–Simpson index or Gini's impurity). Computing quadratic forms has been studied in the context of local DP by Bell et al. [BBGK20] who gave algorithms for functions $f$ that are Lipschitz and for estimating the AUC of a classifier; the latter was improved recently [CM23]. While these problems are simple in central DP (as we can directly add noise to the function value), we note that a clear challenge for computing pairwise statistics (and hence quadratic forms) in local DP is that each summand depends on data points held by two different users.

**Linear Queries.** Computation of quadratic forms is a natural "degree-2" variant of the so-called *linear queries* (defined below), a well-studied problem in the DP literature.

**Definition 2** (Linear Queries). *Given a* workload matrix $W \in \mathbb{R}^{k \times k}$, *the goal is to compute* $W h_{\mathbf{x}}$, *where* $h_{\mathbf{x}} \in \mathbb{Z}_{\geq 0}^k$ *denotes the normalized histogram of the input.*

There is a long line of work on privately answering linear queries. In both the central and local models, the optimal errors achievable by DP algorithms are (mostly) well-understood for various types of errors, such as the $\ell_2^2$-error or the $\ell_\infty$-error [HT10, BDKT12, LMH+15, NTZ16, BUV18, BBNS19, ENU20, Nik23]. Recall that the *MSE* of an estimate $\hat{z}$ of a real value $z$ is defined as $\mathrm{MSE}(\hat{z}, z) := \mathbb{E}_{\hat{z}}[(\hat{z} - z)^2]$. For our purpose, we consider the following natural measure of accuracy:

**Definition 3** (mMSE). *Given a mechanism $\mathcal{M}$ that answers linear queries for a workload matrix $W \in \mathbb{R}^{k \times k}$, its* maximum mean square error (mMSE) *is defined as* $\mathrm{mMSE}(\mathcal{M}; W) := \max_{\mathbf{x} \in [k]^n} \max_{j \in [k]} \mathrm{MSE}(\mathcal{M}(\mathbf{x})_j, (W h_{\mathbf{x}})_j)$.

In a recent seminal work, Edmonds et al. [ENU20] provide a nearly optimal characterization of the error achievable by non-interactive $\varepsilon$-local DP algorithms. To describe their result, we need some additional definitions of norms on matrices and related quantities:

▷ $1 \to 2$ norm: For any matrix $A$, $\|A\|_{1 \to 2}$ is the maximum ($\ell_2$-)norm of its columns.
▷ $\ell_\infty$-norm: For any matrix $A$, $\|A\|_\infty$ is the maximum absolute value among its entries.
▷ Factorization norm: For $W \in \mathbb{R}^{k \times k}$, let $\gamma_2(W) := \min_{L^T R = W} \|L\|_{1 \to 2} \|R\|_{1 \to 2}$.
▷ Approximate-Factorization norm[1]: For $W \in \mathbb{R}^{k \times k}$ and $\alpha \geq 0$, let $\gamma_2(W; \alpha) := \min_{\|\tilde{W} - W\|_\infty \leq \alpha} \gamma_2(\tilde{W})$.

Finally, we define $\zeta(W, n) := \min_{\alpha \geq 0} (\gamma_2(W; \alpha) + \alpha \varepsilon \sqrt{n})$. (Note that $\zeta(W, n) \leq \gamma_2(W)$ as we can pick $\alpha = 0$.) The result of Edmonds et al. [ENU20] states that this quantity, up to polylogarithmic factors, governs the best error achievable for linear queries in the non-interactive local DP model[2]:

**Theorem 4** ([ENU20]). *For any workload matrix $W$, there is a non-interactive $\varepsilon$-local DP mechanism for linear queries with* $\mathrm{mMSE}$ *at most* $O\left(\frac{\zeta(W,n)^2}{\varepsilon^2 n}\right)$. *Furthermore, any non-interactive $\varepsilon$-local DP mechanism must incur* $\mathrm{mMSE}$ *at least* $\tilde{\Omega}\left(\frac{\zeta(W,n)^2}{\varepsilon^2 n}\right)$.

Given such a tight and generic characterization for linear queries, it is natural to ask:

*Can we characterize the error of computing the quadratic form for any matrix $W$ with local DP?*

---

[1]Note that, unlike the three quantities above, this is not actually a norm.

[2]We remark that [ENU20] did not state their bounds in the form we present in Theorem 4 (or even for mMSE); we provide more detail on how to interpret their lower bound in this form in Appendix B. The upper bound is via the *matrix mechanism*, introduced in central DP earlier in [LMH+15, NTZ16].

## 1.1 Our Contributions

In this work, we answer this question by establishing connections between the problems of computing quadratic forms and linear queries. Leveraging the wealth of knowledge on the latter, we obtain several new (and general) upper and lower bounds for the former.

**Non-Interactive Local DP.** First, in the non-interactive setting, we give the following upper and lower bounds. Together, they show that the error one can expect for computing quadratic forms is essentially the same as that for computing linear queries (as provided in Theorem 4).

**Theorem 5** (Non-interactive Algorithm). *For any $W \in \mathbb{R}^{k \times k}$, there is a non-interactive $\varepsilon$-local DP mechanism for estimating quadratic forms on $W$ with $\mathrm{MSE}$ at most $O\left(\frac{\zeta(W,n)^2(\log k)}{\varepsilon^2 n}\right)$.*

**Theorem 6** (Non-interactive Lower Bound). *For any symmetric $W \in \mathbb{R}^{k \times k}$, any non-interactive $\varepsilon$-local DP mechanism for estimating quadratic forms on $W$ must incur $\mathrm{MSE}$ at least $\tilde{\Omega}\left(\frac{\zeta(W,n)^2}{\varepsilon^2 n}\right)$.*

Our results above are generic and can be applied to any pairwise statistics. To demonstrate the power of the algorithm, we now state implications for several classes of statistics that have been studied in the privacy literature. Due to space constraints, we defer their definitions to Section 5.

**Corollary 7.** *There is a non-interactive $\varepsilon$-local DP mechanism for computing pairwise statistics for:*

▷ *Any $G$-Lipschitz function $f : [0, 1] \to \mathbb{R}$, with $\mathrm{MSE}\ O\left(\frac{G^2 \log n}{\varepsilon^2 n}\right)$,*

▷ *Kendall's $\tau$ coefficient, with $\mathrm{MSE}\ O\left(\frac{(\log k)^5}{\varepsilon^2 n}\right)$,*

▷ *AUC under the balancedness assumption[3], with $\mathrm{MSE}\ O\left(\frac{(\log k)^3}{\varepsilon^2 n}\right)$,*

▷ *Gini's diversity index, with $\mathrm{MSE}\ O\left(\frac{\log k}{\varepsilon^2 n}\right)$.*

For $O(1)$-Lipschitz functions, which includes several well-known metrics such as Gini mean difference, we improve upon the algorithm of Bell et al. [BBGK20] whose MSE is $O\left(\frac{1}{\varepsilon\sqrt{n}}\right)$. We are not aware of any previous bounds on Kendall's $\tau$ coefficient before; the metric was mentioned in [BBGK20] without any error guarantee given. For AUC, our bound is the same as in [BBGK20] but worse than a follow-up work [CM23] by a $\log k$ factor; nonetheless, we stress that our result is derived as a corollary of a generic algorithm without relying too deeply on AUC (except for the $\gamma_2$-norm of its matrix). For Gini's diversity index, our bound is again worse than that from Bravo et al. [BHBFG+22] by $\log k$ factor, but their algorithm requires the use of public randomness; we do not use any public randomness. (Throughout this work, when we refer to the local model without further specification, we assume no public randomness, aka *private-coin* model.)

**Remark.** Since our non-interactive algorithm only requires a vector-summation primitive, we can also apply protocols for vector summation in the shuffle model (e.g., [CSU+19, BBGN20, GMPV20]) to obtain an $(\varepsilon, \delta)$-DP protocol in the shuffle model, reducing the MSE by a factor of $n$ for each setting in Theorem 5 and Corollary 7.

**Interactive Local DP.** Finally, we also provide an interactive algorithm whose MSE does not depend on $\gamma_2(W)$, as long as $n$ is sufficiently large:

**Theorem 8** (Interactive Algorithm). *For any $W \in \mathbb{R}^{k \times k}$, there is a three-round $\varepsilon$-DP algorithm for estimating quadratic forms on $W$ such that, for $n \geq \left(\frac{\gamma_2(W) \cdot \log k}{\|W\|_\infty \cdot \varepsilon}\right)^{O(1)}$, the $\mathrm{MSE}$ is $O\left(\frac{\|W\|_\infty^2}{\varepsilon^2 n}\right)$.*

We note that the dependence $O\left(\frac{\|W\|_\infty^2}{\varepsilon^2 n}\right)$ is the best possible: even when the $k = 2$ and $W$ is binary, the problem is as hard as binary summation, which is known to require MSE at least $\tilde{\Omega}\left(\frac{1}{\varepsilon^2 n}\right)$ even for an arbitrary number of rounds of interactions [CSS12], and we can rescale this hard instance to get any desired $\ell_\infty$-norm.

---

[3]The balancedness assumption for AUC states that there are $\Omega(n)$ examples with each label 0, 1. This is a required assumption to achieve an error in the form presented, as otherwise when e.g., there is a single 0-labeled example, it is impossible to achieve any non-trivial guarantee.

Combining the non-interactive lower bound in Theorem 6 and the interactive upper bound in Theorem 8, our results imply that there exists a matrix $W \in \mathbb{R}^{k \times k}, \varepsilon > 0, n \in \mathbb{N}$ such that interactive algorithms are provably more accurate than non-interactive ones for privately computing the quadratic form on $W$. This places computing pairwise statistics as one of the few problems (and perhaps the most natural) that are known to separate interactive and non-interactive local DP. Due to space constraints, we defer further discussion of this to the Appendix.

## 1.2 Technical Overview

We now give a brief technical overview of our proofs.

**Black-Box Reduction from Linear Queries (Lower Bound).** As mentioned earlier, our results are shown through connections between computing quadratic forms and linear queries. We will start with black-box reductions between the two. Suppose that we have a non-interactive $\varepsilon$-local DP algorithm $\mathbb{A}$ for computing the quadratic form on $W$. We will construct a non-interactive $\varepsilon$-local DP algorithm $\mathbb{A}'$ for computing linear queries of $W$. This will allow us to establish our lower bound (Theorem 6) as a consequence of the lower bound for linear queries (Theorem 4).

For simplicity of this overview, instead of considering the full quadratic form, suppose that $\mathbb{A}$ works on $2n$ users with inputs $x_1, \ldots, x_n, y_1, \ldots, y_n$ and computes an estimate of $h_{\mathbf{y}}^T W h_{\mathbf{x}}$. On input $x_1, \ldots, x_n$, algorithm $\mathbb{A}$ works as follows:

▷ Each user $i$ runs the randomizer of $\mathbb{A}$ on $x_i$ to get a response $o_i$ and sends it to the analyst.
▷ For each $j \in [k]$, the analyst *simulates* running the randomizer of $\mathbb{A}$ on $y_1 = \cdots = y_n = j$ to get responses $o'_1, \ldots, o'_n$. Then, the analyst lets $\hat{z}_j$ be the estimator of $\mathbb{A}$ based on the responses $o_1, \ldots, o_n, o'_1, \ldots, o'_n$.
▷ The analyst then outputs $(\hat{z}_1, \ldots, \hat{z}_k)$ as its estimate.

In other words, the randomized responses from $\mathbb{A}$ are used as an "oracle" for $\mathbb{A}'$ to compute the different linear queries. The key observation here is that, when we set $y_1 = \cdots = y_n = j$, $\mathbb{A}$ produces an estimate for $h_{\mathbf{y}}^T W h_{\mathbf{x}} = \mathbf{1}_j^T W h_{\mathbf{x}} = (W h_{\mathbf{x}})_j$ as desired. Note that this reduction only works in the non-interactive setting: if we were in the interactive setting, the responses on $x_1, \ldots, x_n$ (of protocol $\mathbb{A}$) would have been dependent on those of $y_1, \ldots, y_n$. Therefore, the first step of the reduction would have been impossible.

While this encapsulates the high-level ideas of our proof, there are some details that needs to be handled. E.g., if $\mathbb{A}$ outputs the quadratic form $h_{\mathbf{x} \cup \mathbf{y}}^T W h_{\mathbf{x} \cup \mathbf{y}}$, there are "cross terms" of the form $h_{\mathbf{x}}^T W h_{\mathbf{x}}$ that need to be removed. We formally describe and analyze the full reduction in Section 4.

**Black-Box Reduction to Linear Queries (Algorithm).** We can also give a reduction in the reverse direction, although this results in an additional round of communication. Specifically, let $\mathbb{A}'$ be a non-interactive $\varepsilon$-local DP algorithm for computing linear queries of $W$. We can construct a two-round $(2\varepsilon)$-local DP algorithm $\mathbb{A}$ for computing the quadratic form on $W$ as follows.

▷ **First Round:** Run $\mathbb{A}'$ on all users to compute an estimate $(\hat{z}_1, \ldots, \hat{z}_k)$ for $W h_{\mathbf{x}}$.
▷ **Second Round:** Each user $j \in [n]$, sends $o_j = \hat{z}_{x_j} + \kappa_j$ to the analyst where $\kappa_j$ is (appropriately calibrated) Laplace noise. The analyst then outputs $\frac{1}{n}(o_1 + \cdots + o_n)$.

Again, we omit some details for simplicity, such as the fact that each $\hat{z}_j$ may not be bounded a priori, which may make the second step violate DP. However, these are relatively straightforward to handle.

If there were no noise, we would have $o_j = \mathbf{1}_{x_j}^T W h_{\mathbf{x}}$ and thus $\frac{1}{n}(o_1 + \cdots + o_n) = h_{\mathbf{x}}^T W h_{\mathbf{x}}$ as desired. Furthermore, it is not hard to see that the error from $\kappa_1, \ldots, \kappa_n$ is dominated by the error from $\mathbb{A}'$ in the first step. In other words, we get an error similar to the one in Theorem 5 here, but the protocol is interactive (two-round). Making the protocol non-interactive requires us to step away from the black-box approach and open up the linear query algorithm (Theorem 4).

**White-Box Algorithms.** For simplicity, in this section we describe protocols with accuracy that depends on $\gamma_2(W)$, which can be larger than $\zeta(W, n)$. The desired error dependence on $\zeta(W, n)$ can be obtained from this bound via a reduction, as shown in Lemma 10.

To understand our algorithm, we first describe the *matrix mechanism* for linear queries (cf. [ENU20]). That algorithm works as follows: factorize $W = L^T R$. Then, user $i$ sends $o_i^R = R\mathbf{1}_{x_i} + z_i^R$

where $z_i^R$ is a appropriately selected random noise. In other words, each user privatizes $W\mathbf{1}_{x_i}$ and sends it to the analyst. Finally, the analyst outputs $L^T\left(\frac{1}{n}(o_1^R + \cdots + o_n^R)\right)$. This is exactly equal to $Wh_\mathbf{x} + L^T Z^R$ where $Z^R := \frac{1}{n}(z_1^R + \cdots + z_n^R)$. It is possible to select the noise in such a way that $Z^R$ is $\left(\|R\|_{1\to 2}/\varepsilon\sqrt{n}\right)$-sub-Gaussian. This leads to an mMSE of $O\left(\frac{\gamma_2(W)^2}{\varepsilon^2 n}\right)$.

This suggests a natural approach for quadratic forms: in addition to sending (a privatized version of) $R\mathbf{1}_{x_i}$, the user sends a privatized version of $L\mathbf{1}_{x_i}$, i.e., $o_i^L = L\mathbf{1}_{x_i} + z_i^L$ where $z_i^L$ is a random noise, to the analyst. The analyst then outputs $\left\langle\frac{1}{n}(o_1^L + \cdots + o_n^L), \frac{1}{n}(o_1^R + \cdots + o_n^R)\right\rangle$. Letting $Z^L := \frac{1}{n}(z_1^L + \cdots + z_n^L)$, the output can be written as $h_\mathbf{x}^T W h_\mathbf{x} + \langle Lh_\mathbf{x}, Z^R\rangle + \langle Rh_\mathbf{x}, Z^L\rangle + \langle Z^L, Z^R\rangle$. There are three error terms: $\langle Lh_\mathbf{x}, Z^R\rangle, \langle Rh_\mathbf{x}, Z^L\rangle$, and $\langle Z^L, Z^R\rangle$. Similar to linear queries analysis, it is not hard to see that the first two terms contribute $O\left(\frac{\gamma_2(W)^2}{\varepsilon^2 n}\right)$ to the MSE. Unfortunately, the last term is problematic for us: a simple calculation shows that it contributes $O\left(\frac{\ell}{\varepsilon^4 n^2}\right)$ to the MSE, where $\ell$ denotes the number of rows of $L, R$. A priori, this term can be quite large as there is no obvious bound on $\ell$. In fact, if $W$ is full rank (which is the case for most popular pairwise statistics), then we know that $\ell$ must be at least $k$. This leads to an undesired error term $O\left(\frac{k}{\varepsilon^4 n^2}\right)$, which dominates the first term for small-to-moderate values of $n$, i.e., when $n \leq \frac{k}{\varepsilon^2\gamma_2(W)^2}$.

To overcome this, we observe that, if we only look for *approximate factorization* (in the same sense as $\gamma_2(W;\alpha)$ defined above), then it is always possible to reduce $\ell$ via dimensionality reduction techniques (e.g., [DG03]). Namely, we may pick a (e.g., random Gaussian) matrix $A$ and replace $L, R$ with $AL, AR$ respectively. Selecting the number of rows of $A$ appropriately then yields Theorem 5.

**Additional Interactive Algorithms.** To describe our algorithm, let us start by instantiating the above black-box two-round reduction using the matrix mechanism. In this context, the reduction yields the following algorithm:

▷ **First Round:** Each user $i$ sends $o_i^L = L\mathbf{1}_{x_i} + z_i^L$ to the analyst, where $z_i^L$ is appropriately selected random noise. The analyst then broadcasts $O^L = \frac{1}{n}(o_1^L + \cdots + o_n^L)$ to each user.

▷ **Second Round:** Each user $j \in [n]$ sends $o_j = \langle O^L, R\mathbf{1}_{x_j}\rangle + \kappa_j$ to the analyst, where $\kappa_j$ is appropriately calibrated Laplace noise. The analyst then outputs $\frac{1}{n}(o_1 + \cdots + o_n)$.

It is not hard to see that the $\kappa_j$ noise terms together contribute at most $O(1/\varepsilon^2 n)$ to the MSE. Therefore, the main noise comes from the first step. Similar to the previous discussion, this noise can be written as $\langle Rh_\mathbf{x}, Z^L\rangle$ where $Z^L := \frac{1}{n}(z_1^L + \cdots + z_n^L)$. The contribution of this noise to the MSE is then $O\left(\frac{\gamma_2(W)^2}{\varepsilon^2 n}\right)$. The $\gamma_2^2(W)$ term shows up in the error because $\|Rh_\mathbf{x}\|_2$ can be as large as $\|R\|_{1\to 2}$. The idea motivating our improvement is simple: Can we replace the $Rh_\mathbf{x}$ term with a term that is much smaller?

This brings us to the following strategy. We will use an additional round at the beginning to compute a rough estimate $\mu$ of $Rh_\mathbf{x}$. Using the so-called *projection mechanism* [BBNS19][4], it is possible to compute $\mu$ such that $\|Rh_\mathbf{x} - \mu\| \leq \frac{(\gamma_2(W)\cdot(\log k)/\varepsilon)^{O(1)}}{n^{\Omega(1)}}$. The subspace orthogonal to $\mu$ can be processed in a similar manner as before, but now the error term will just be $\langle Rh_\mathbf{x} - \mu, Z^L\rangle$. Since $\|Rh_\mathbf{x} - \mu\|$ is now much smaller (approaches 0 as $n \to \infty$), this gives us the improved error bound. We note that the direction of $\mu$ can be handled by having each user $i$ directly send $\langle o_i^L, \mu\rangle$ plus an appropriately calibrated noise. This summarizes the high-level idea of our approach.

Due to space constraints, we focus on the non-interactive algorithms in the main body and defer the proof of the interactive algorithm to the Appendix.

## 2 Preliminaries

**Differential Privacy.** For $\varepsilon \geq 0$, an algorithm $\mathcal{M}$ is $\epsilon$-*DP* if for every pair $X, X'$ of inputs that differ on one user's input and for every possible output $o$, $\Pr[\mathcal{M}(X) = o] \leq e^\varepsilon \cdot \Pr[\mathcal{M}(X') = o]$.

---

[4]See also [NTZ16] for the original projection mechanism that was proposed for the central model.

An algorithm $\mathbb{A}$ in the local DP model consists of a randomizer $\mathcal{R}$ and an analyst, computed as $\mathbb{A}(X) = \text{analyst}(\mathcal{R}(x_1), \ldots, \mathcal{R}(x_n))$ for input $X = \{x_1, \ldots, x_n\}$. $\mathbb{A}$ is said to be (non-interactive) $\varepsilon$-*local DP* if $\mathcal{R}$ is $\varepsilon$-DP.

A real-valued random variable $Z$ is $\sigma$-*sub-Gaussian* iff $\mathbb{E}[\exp(Z^2/\sigma^2)] \leq 2$. A $\mathbb{R}^d$-valued random variable $Z$ is $\sigma$-sub-Gaussian iff $\langle \theta, Z \rangle$ is $\sigma$-sub-Gaussian for all unit vectors $\theta \in \mathbb{R}^d$.

**Theorem 9** ([BBNS19]). *For any $C, \varepsilon > 0$, there is a (non-interactive) $\varepsilon$-local DP algorithm* $\text{VRand}_{\varepsilon,C}$ *that takes in $x \in \mathbb{R}^d$ such that $\|x\|_2 \leq C$ and outputs $Y \in \mathbb{R}^d$ such that $\mathbb{E}[Y] = x$ and $Y - x$ is $\sigma$-sub-Gaussian for $\sigma = O(C/\varepsilon)$.*

**Error: Factorization vs Approximate-Factorization.** We show that it suffices to give errors in terms of $\gamma_2(W)$ instead of $\zeta(W; n)$; this will be convenient for our subsequent proofs. Due to space constraints, the proof of the following statement is deferred to Appendix A.

**Lemma 10.** *Suppose that, for all $W \in \mathbb{R}^{k \times k}$, there is a non-interactive $\varepsilon$-local DP protocol $\mathbb{A}$ for quadratic form on $W$ with* $\text{MSE}$ $O\left(c(n, \varepsilon, k) \cdot \frac{\gamma_2(W)^2}{\varepsilon^2 n}\right)$ *where $c(n, \varepsilon, k) \geq \Omega(1)$. Then there is also a non-interactive $\varepsilon$-local DP protocol $\mathbb{A}'$ with* $\text{MSE}$ $O\left(c(n, \varepsilon, k) \cdot \frac{\zeta(W; n)^2}{\varepsilon^2 n}\right)$.

## 3 Non-Interactive Algorithm

In this section, we prove Theorem 5. To do so, let us start by defining (approximate) *rank-restricted factorization norms*[5], which are the same as $\gamma_2(W), \gamma_2(W; \alpha)$ except we now restrict the number of rows of $L, R$ to be at most $\ell$:

▷ For $W \in \mathbb{R}^{k \times k}, \ell \in \mathbb{N}$, let $\gamma_2^\ell(W) := \min_{L^T R = W; L, R \in \mathbb{R}^{\ell \times k}} \|L\|_{1 \to 2} \|R\|_{1 \to 2}$.

▷ For $W \in \mathbb{R}^{k \times k}, \ell \in \mathbb{N}$ and $\alpha \geq 0$, let $\gamma_2^\ell(W; \alpha) := \min_{\|\tilde{W} - W\|_\infty \leq \alpha} \gamma_2^\ell(\tilde{W})$.

We can now use the approximate rank-restricted factorization to perform the algorithm as outlined in Section 1.2 with an error term that depends on $\ell$ (and $\alpha$):

**Lemma 11.** *For any $W \in \mathbb{R}^{k \times k}, \ell \in \mathbb{N}$, and $\alpha \geq 0$, there is a non-interactive $\varepsilon$-local DP algorithm that estimates the quadratic form on $W$ to within an MSE of $O\left(\alpha^2 + \gamma_2^\ell(W; \alpha)^2 \cdot \left(\frac{1}{\varepsilon^2 n} + \frac{\ell}{\varepsilon^4 n^2}\right)\right)$.*

*Proof.* By definition of $\gamma_2^\ell(W; \alpha)$, there exists $\tilde{W} \in \mathbb{R}^{k \times k}, L, R \in \mathbb{R}^{\ell \times k}$ such that $\|\tilde{W} - W\|_\infty \leq \alpha$ and $\tilde{W} = L^T R$ where, w.l.o.g., by rescaling, $\|L\|_{1 \to 2} = \|R\|_{1 \to 2} = \sqrt{\gamma_2^\ell(W)} =: C$.

**Algorithm Description.** Let $\bar{\varepsilon} = \varepsilon/2$. The algorithm works as follows:

▷ Each user $i \in [n]$ sends $y_i^L \leftarrow \text{VRand}_{\bar{\varepsilon},C}(L\mathbf{1}_{x_i})$ and $y_i^R \leftarrow \text{VRand}_{\bar{\varepsilon},C}(R\mathbf{1}_{x_i})$ to the analyst (where $\text{VRand}.(\cdot)$ is from Theorem 9).

▷ The analyst outputs $\left\langle \frac{1}{n} \sum_{i \in [n]} y_i^L, \frac{1}{n} \sum_{i \in [n]} y_i^R \right\rangle$.

As each user runs an $(\varepsilon/2)$-DP randomizer twice, the algorithm is $\varepsilon$-DP.

**Utility Analysis.** Let $z_i^L = y_i^L - L\mathbf{1}_{x_i}$ and $z_i^R = y_i^R - R\mathbf{1}_{x_i}$. From Theorem 9, $z_i^L, z_i^R$ are zero-mean and $\sigma$-sub-Gaussian for $\sigma = O(C/\varepsilon)$. Let $Z^L := \frac{1}{n} \sum_{i \in [n]} z_i^L, Z^R := \frac{1}{n} \sum_{i \in [n]} z_i^R$; we then have that these are zero-mean and $\sigma'$-sub-Gaussian for $\sigma' = \sigma/\sqrt{n}$. The MSE of the protocol is given by

$$
\mathbb{E}\left[\left(\left\langle \frac{1}{n} \sum_{i \in [n]} y_i^L, \frac{1}{n} \sum_{i \in [n]} y_i^R \right\rangle - h_{\mathbf{x}}^T W h_{\mathbf{x}}\right)^2\right]
$$

$$
= \mathbb{E}\left[\left(h_{\mathbf{x}}^T(\tilde{W} - W)h_{\mathbf{x}} + \left\langle Z^L, Rh \right\rangle + \left\langle Lh, Z^R \right\rangle + \left\langle Z^L, Z^R \right\rangle\right)^2\right]
$$

$$
\lesssim \mathbb{E}(h_{\mathbf{x}}^T(\tilde{W} - W)h_{\mathbf{x}})^2 + \mathbb{E}\left\langle Z^L, Rh \right\rangle^2 + \mathbb{E}\left\langle Lh, Z^R \right\rangle^2 + \mathbb{E}\left\langle Z^L, Z^R \right\rangle^2
$$

$$
\lesssim \|\tilde{W} - W\|_\infty^2 + (\sigma')^2 \|Rh\|_2^2 + (\sigma')^2 \|Lh\|_2^2 + \ell \cdot (\sigma')^4
$$

---

[5]These are not actually norms, but we use the term for consistency with other similar quantities.

$$\leq \alpha^2 + (\sigma')^2\|R\|_{1\to2}^2 + (\sigma')^2\|L\|_{1\to2}^2 + \ell \cdot (\sigma')^4$$
$$\lesssim \alpha^2 + (\sigma')^2 C^2 + \ell \cdot (\sigma')^4$$
$$\lesssim \alpha^2 + \frac{C^4}{\varepsilon^2 n} + \frac{\ell \cdot C^4}{\varepsilon^4 n^2}. \qquad \square$$

**Rank-Restricted Approximate Factorization via JL.** We next show that w.l.o.g. we can take $\ell$ to be quite small in Lemma 11 above.

**Lemma 12.** *Let $W \in \mathbb{R}^{k\times k}$ and $\alpha \in (0, \gamma_2(W))$, there is $\ell \lesssim \frac{\gamma_2(W)^2 \log k}{\alpha^2}$ with $\gamma_2^\ell(W; \alpha) \lesssim \gamma_2(W)$.*

As mentioned earlier, this lemma is proved by applying Johnson–Lindenstrauss (JL) dimensionality reduction to each column of $L, R$. We summarize a simplified version of the JL lemma below.

**Lemma 13** (Johnson–Lindenstrauss Lemma (e.g., [DG03])). *Let $\beta \in (0,1)$ and $U = \{u_1, \ldots u_m\} \subseteq \mathbb{R}^d$. For some $\ell \leq O(\beta^{-2} \log m)$, there exists a matrix $A \in \mathbb{R}^{\ell\times d}$ such that for all $u, v \in U$, we have $(1-\beta)\|u-v\|_2^2 \leq \|Au - Av\|_2^2 \leq (1+\beta)\|u-v\|_2^2$.*

We are now ready to prove Lemma 12.

*Proof of Lemma 12.* By definition of $\gamma_2(W)$, there exists $L, R \in \mathbb{R}^{d\times k}$ for some $d \in \mathbb{N}$ such that $W = L^T R$ where$\|L\|_{1\to2} = \|R\|_{1\to2} = \sqrt{\gamma_2(W)} =: C$.

Let $L_1, \ldots, L_k$ (resp. $R_1, \ldots, R_k$) be the columns of $L$ (resp. $R$). Consider $U = \{0, L_1, \ldots, L_k, R_1, \ldots, R_k, -L_1, \ldots, -L_k, -R_1, \ldots, -R_k\}$ and $\beta = 0.5\alpha/C^2$; let $\ell = O\left(\beta^{-2}\log k\right) = O\left(\frac{\gamma_2(W)^2 \log k}{\alpha^2}\right)$ and $A \in \mathbb{R}^{\ell\times d}$ be as guaranteed by Lemma 13.

Let $\tilde{L} = AL, \tilde{R} = AR$, and $\tilde{W} = \tilde{L}^T\tilde{R}$. For all $i \in [k]$, we have $\|\tilde{L}_i\|_2^2 \leq (1+\beta)\|L_i\|_2^2$ and $\|\tilde{R}_i\|_2^2 \leq (1+\beta)\|R_i\|_2^2$. Therefore, $\|\tilde{L}\|_{1\to2}, \|\tilde{R}\|_{1\to2} \leq O(C)$, i.e., $\gamma_2^\ell(\tilde{W}) \leq O(\gamma_2(W))$. Moreover, for each $i, j \in [k]$, we have

$$\left|\tilde{W}_{i,j} - W_{i,j}\right| = \left|\left\langle \tilde{L}_i, \tilde{R}_j \right\rangle - \langle L_i, R_j \rangle\right|$$
$$\leq \frac{1}{4}\left(\left|\|\tilde{L}_i + \tilde{R}_j\|_2^2 - \|L_i + R_j\|_2^2\right| + \left|\|\tilde{L}_i - \tilde{R}_j\|_2^2 - \|L_i - R_j\|_2^2\right|\right)$$
$$\leq \frac{\beta}{4}\left(\|L_i + R_j\|_2^2 + \|L_i - R_j\|_2^2\right)$$
$$\leq 2\beta C^2 \leq \alpha.$$

Thus, $\|\tilde{W} - W\|_\infty \leq \alpha$ and therefore $\gamma_2^\ell(W; \alpha) \leq O(\gamma_2(W))$ as claimed. $\qquad\square$

We end this section by proving Theorem 5, which is a simple combination of Lemma 11 and Lemma 12.

*Proof of Theorem 5.* Pick[6] $\alpha = \frac{\gamma_2(W)}{\varepsilon\sqrt{n}}$ and apply Lemma 12: There exists $\ell = O\left(\frac{\gamma_2(W)^2 \log k}{\alpha^2}\right) = O\left((\log k)\varepsilon^2 n\right)$ such that $\gamma_2^\ell(W; \alpha) \leq O(\gamma_2(W))$. Plugging this back into Lemma 11 then gives us a non-interactive $\varepsilon$-local DP protocol with MSE

$$\lesssim \alpha^2 + \gamma_2^\ell(W;\alpha)^2 \cdot \left(\frac{1}{\varepsilon^2 n} + \frac{\ell}{\varepsilon^4 n^2}\right) \lesssim \gamma_2(W)^2 \left(\frac{1}{\varepsilon^2 n} + \frac{\log k}{\varepsilon^2 n}\right) \lesssim \frac{\gamma_2(W)^2 \log k}{\varepsilon^2 n}.$$

Applying Lemma 10 then concludes the proof. $\qquad\square$

## 4 Lower Bounds for Non-Interactive Algorithms

In this section we formalize the reduction from linear queries to computing quadratic forms, as outlined in Section 1.2. The properties of the reduction are stated in the theorem below.

---

[6]We may assume w.l.o.g. that $\alpha \leq \gamma_2(W)$; otherwise, the guarantee in Theorem 5 is trivial, i.e., always outputting 0 satisfies the bound.

**Theorem 14.** *Let $W \in \mathbb{R}^{k \times k}$ be symmetric. Suppose that there is a non-interactive $\varepsilon$-local DP mechanism for computing the quadratic form on $W$ with MSE at most $\alpha(\varepsilon, n)$. Then there is a non-interactive $\varepsilon$-local DP protocol for computing the linear queries of $W$ with mMSE $O(\alpha(\varepsilon/2, n) + \alpha(\varepsilon/2, 2n))$.*

Theorem 14 and the lower bound in Theorem 4 immediately imply Theorem 6. In the proof below, we use subscripts $\varepsilon, n$ to denote the privacy loss parameter and the number of users in the protocol.

*Proof of Theorem 14.* Let $\mathbb{A}_{\varepsilon,n}$ be the $\varepsilon$-local DP protocol for computing the quadratic form on $W$, and let $\mathcal{R}_{\varepsilon,n}$ denote its randomizer. We construct an algorithm $\mathbb{A}'_{\varepsilon,n}$ for linear queries with workload matrix $W$. On input $x_1, \ldots, x_n$, proceed as follows:

▷ Run the protocol of $\mathbb{A}_{\varepsilon/2,n}$ on $x_1, \ldots, x_n$ to compute an estimate $\hat{z}$ for $h_{\mathbf{x}}^T \mathbb{A} h_{\mathbf{x}}$.

▷ In the same round as above, run the randomizer $\mathcal{R}_{\varepsilon/2,2n}$ of $\mathbb{A}_{\varepsilon/2,2n}$ on $x_1, \ldots, x_n$ to get the responses $\mathcal{R}_{\varepsilon/2,2n}(x_1), \ldots, \mathcal{R}_{\varepsilon/2,2n}(x_n)$.

▷ For each $j \in [k]$, do the following:

   ▷ Run the randomizer $\mathcal{R}$ of $\mathbb{A}$ on $y_1 = \cdots = y_n = j$ to get the responses $\mathcal{R}_{\varepsilon/2,2n}(y_1), \ldots, \mathcal{R}_{\varepsilon/2,2n}(y_n)$.

   ▷ Compute the estimator $z'_j$ of $\mathbb{A}$ on the $2n$ responses $\mathcal{R}_{\varepsilon/2,2n}(x_1), \ldots, \mathcal{R}_{\varepsilon/2,2n}(x_n)$, $\mathcal{R}_{\varepsilon/2,2n}(y_1), \ldots, \mathcal{R}_{\varepsilon/2,2n}(y_n)$.

   ▷ Set $\hat{z}_j = 2z'_j - \frac{1}{2}\hat{z} - \frac{1}{2}\mathbf{1}_j^T W \mathbf{1}_j$.

▷ Output $(\hat{z}_1, \ldots, \hat{z}_j)$ as the estimates for the linear queries.

Since $(\varepsilon/2)$-local DP randomizers are run on each input $x_i$ twice, the basic composition theorem implies that this is a $\varepsilon$-local DP algorithm as desired.

For each $j \in [k]$, we now compute the MSE of $\hat{z}_j$. First, observe that

$$(Wh_{\mathbf{x}})_j = \mathbf{1}_j^T W h_{\mathbf{x}} = 2\left(h_{\mathbf{x}\cup\mathbf{y}}^T W h_{\mathbf{x}\cup\mathbf{y}}\right) - \frac{1}{2}h_{\mathbf{x}}^T W h_{\mathbf{x}}^T - \frac{1}{2}\mathbf{1}_j^T W \mathbf{1}_j,$$

where $\mathbf{y}$ denotes the dataset with $n$ copies of $j$.

Therefore, we can bound the MSE of $\hat{z}_j$ as follows:

$$
\begin{aligned}
\mathrm{MSE}(\hat{z}_j; (Wh_{\mathbf{x}})_j) &= \mathbb{E}\left[(\hat{z}_j - (Wh_{\mathbf{x}})_j)^2\right] \\
&= \mathbb{E}\left[\left(2\left(z'_j - h_{\mathbf{x}\cup\mathbf{y}}^T W h_{\mathbf{x}\cup\mathbf{y}}\right) + \frac{1}{2}\left(\hat{z} - h_{\mathbf{x}}^T W h_{\mathbf{x}}^T\right)\right)^2\right] \\
&\lesssim \mathbb{E}\left[\left(z'_j - h_{\mathbf{x}\cup\mathbf{y}}^T W h_{\mathbf{x}\cup\mathbf{y}}\right)^2\right] + \mathbb{E}\left[\left(\hat{z} - h_{\mathbf{x}}^T W h_{\mathbf{x}}^T\right)^2\right] \\
&\leq \alpha(\varepsilon/2, 2n) + \alpha(\varepsilon/2, n),
\end{aligned}
$$

where the last inequality follows from the guarantees of $\mathbb{A}$. Thus, the mMSE of $\mathbb{A}'$ is at most $O(\alpha(\varepsilon/2, 2n) + \alpha(\varepsilon/2, n))$, as desired. $\qquad\square$

## 5 Upper Bounds for Specific Metrics

In this section, we obtain concrete upper bounds for many well-known U-statistics of degree 2. For a kernel $f : \mathcal{X} \to \mathbb{R}$, let $W^f \in \mathbb{R}^{\mathcal{X} \times \mathcal{X}}$ denote the matrix defined by $W_{x,x'}^f = f(x, x')$. Our proof of Corollary 7 proceeds by providing an upper bound on $\gamma_2(W^f)$ for each U-statistic with kernel $f$; the bounds immediately follow from Theorem 5. Similar to before, let $k$ denote $|\mathcal{X}|$.

The following (non-trivial) facts about the factorization norm are useful to keep in mind:

**Fact 15.** *The factorization norm $\gamma_2$ satisfies the following properties:*

   1. *[TJ89] For any $A, B$, we have $\gamma_2(A + B) \leq \gamma_2(A) + \gamma_2(B)$.*
   2. *[LSS08] For any $A, B$, $\gamma_2(A \otimes B) = \gamma_2(A) \cdot \gamma_2(B)$.*

**Gini's diversity index.** For $\mathcal{X} \subseteq \mathbb{R}$, the kernel $f(x, x') = \mathbf{1}[x \neq x']$ captures *Gini's diversity index*. From Fact 15(1), we have $\gamma_2(W^f) \leq \gamma_2(\mathbf{1}_{k \times k}) + \gamma_2(\mathbf{I}_{k \times k}) \leq 1 + 1$ where the inequality $\gamma_2(\mathbf{1}_{k \times k}) \leq 1$ is from the factorization $L = R = \mathbf{1}_k$ and $\gamma_2(\mathbf{I}_{k \times k}) \leq 1$ is from $L = R = \mathbf{I}_{k \times k}$.

**Kendall's $\tau$ coefficient and AUC.** For $\mathcal{X} = A \times B \subseteq \mathbb{R}^2$ with $x_i = (y_i, z_i)$, the kernel $f((y_i, z_i), (y_j, z_j)) = \mathrm{sgn}(y_i - y_j) \cdot \mathrm{sgn}(z_i - z_j)$ yields *Kendall's $\tau$ coefficient*. Let $U_m \in \{-1, 1\}^m$ denote the matrix that has $+1$ on all entries above the main diagonal (inclusive) and $-1$ elsewhere. It is known that $\gamma_2(U_m) = \Theta(\log m)$. We can bound $\gamma_2(W^f)$ for Kendall's $\tau$ coefficient by observing that $W^f = U_A \otimes U_B$. From Fact 15(2), $\gamma_2(W_f) \leq \gamma_2(U_A) \cdot \gamma_2(U_B) \lesssim (\log |A|)(\log |B|) \leq (\log k)^2$.

*AUC* for binary classification is defined in a similar manner as Kendall's tau coefficient, except that (i) $B = \{0, 1\}$ and (ii) the normalization constant being $\frac{1}{n^+ n^-}$ instead if $\frac{1}{\binom{n}{2}}$ where $n^+$ (resp., $n^-$) denotes the number of 1-labeled (resp., 0-labeled) examples. The AUC result follows from the above since $|B| = 2$ in this case. We remark that, for the AUC case, we also have to split the privacy budget and use half of it to estimate $n^+$ so that we can renormalize correctly. It is not hard to see that this renormalization procedure results in at most an additive factor of $O\left(\frac{1}{\varepsilon^2 n}\right)$ in the MSE, under the *balancedness assumption* that $n^-, n^+ \geq \Omega(n)$.

**Lipschitz Losses.** Let $\mathcal{X} = [0, 1]$ and let $f : \mathcal{X} \to \mathbb{R}$ be any function such that $|f(x) - f(x')| \leq G \cdot |x - x'|$; we call $f$ *$G$-Lipschitz*. This class includes U-statistics such as the *Gini's mean difference*, which is given by the kernel $f(x_i, x_j) = |x_i - x_j|$, which is 1-Lipschitz.

Similar to [BBGK20], we use a discrete case where $\mathcal{X} = [k]$ as a subroutine. Our algorithm for this is stated below. Note that Corollary 16 immediately implies the bound for the continuous case: given any function $f : [0, 1] \to \mathbb{R}$, we may select $k$ to be sufficiently large, e.g., $k = \Theta(\varepsilon n^2)$, and discretize the function over the points $1/k, 2/k, \ldots, k/k$. Defining $g : [k] \to \mathbb{R}$ by $g(i) = f(i/k)$ allows us to use Corollary 16 with Lipschitz constant $G/k$. This leads to a MSE of $O\left(\frac{G^2 \log(\varepsilon n^2)}{\varepsilon^2 n}\right)$. The MSE resulting from the discretization error is then at most $\frac{G^2 n^2}{k^2} < \frac{G^2}{\varepsilon^2 n}$.

**Corollary 16** (Discrete Lipschitz)**.** *Assuming $\mathcal{X} = [k]$ and that $f$ is $G$-Lipschitz. There is a non-interactive $\varepsilon$-local DP algorithm for computing pairwise statistics for $f$ with* MSE $O\left(\frac{G^2 k^2 (\log k)}{\varepsilon^2 n}\right)$.

Again, the above corollary follows from Theorem 5 and the following factorization.

**Lemma 17.** *Assuming that $f : [k] \to \mathbb{R}$ is $G$-Lipschitz, then $\gamma_2(W^f) \leq O(Gk)$.*

*Proof.* We assume w.l.o.g. that $k = 2^q - 1$ for some $q \in \mathbb{N}$ and $\|W^f\|_\infty \leq Lk$; otherwise, we may shift $W^f$ (and the answer) without incurring any additional error. We arrange $[k]$ into a balanced binary search tree $\mathcal{T}$ of depth $q - 1$ naturally (where the root is $2^{q-1}$ and the leaves are $1, 3, \ldots, k$). Let $P(j)$ denote the path from node $j$ to the root (inclusive) in $\mathcal{T}$, and let $\ell(j)$ denote the depth of $j$ (where the root has depth 0). Furthermore, let $\mathrm{parent}(j)$ denote the parent of $j$ in $\mathcal{T}$. For notational convenience, let $\mathrm{parent}(2^{q-1}) = \bot$ and let $f(i, \bot) = 0$ for all $i \in [k]$.

We construct $L, R \in \mathbb{R}^{k \times k}$ as follows.

▷ For all $i, j \in [k]$, let $R_{i,j} = \left(\frac{5}{6}\right)^{\ell(i)} \mathbf{1}[i \in P(j)]$.
▷ For all $i, j \in [k]$, $L_{j,i} = \left(\frac{6}{5}\right)^{\ell(j)} (f(i, j) - f(i, \mathrm{parent}(j)))$.

For $i, j \in [k]$, we have $(L^T R)_{i,j} = \sum_{t \in P(j)} (f(i, t) - f(i, \mathrm{parent}(t))) = f(i, j)$. Thus, $L^T R = W$.

Furthermore, $\|R\|_{1 \to 2}^2 = 1 + \left(\frac{5}{6}\right)^2 + \cdots + \left(\frac{5}{6}\right)^{2(q-1)} \lesssim 1$. Meanwhile, we can bound $\|L\|_{1 \to 2}^2$ by

$$\max_{i \in [k]} \sum_{j \in [k]} \left(\left(\frac{6}{5}\right)^{\ell(j)} (f(i, j) - f(i, \mathrm{parent}(j)))\right)^2 \lesssim \sum_{d=0}^{q-1} 2^d \left(\left(\frac{6}{5}\right)^d \cdot \frac{Gk}{2^d}\right)^2 \lesssim G^2 k^2,$$

where the first inequality follows since $f$ is $G$-Lipschitz. Thus $\gamma_2(W^f) \leq O(Gk)$. $\quad\square$

# 6 Conclusion and Open Questions

In this work, we systematically study the problem of privately computing pairwise statistics. We give a non-interactive local DP algorithm and a nearly-matching lower bound for the problem. Furthermore, we show that, for some metrics, improvements can be made if interaction is allowed.

There are several immediate questions from our work. For example, is it possible to remove the $\log k$ multiplicative factor in our non-interactive algorithm (Theorem 5)? Similarly, can the second additive term in our interactive algorithm be removed? As also suggested by [BBGK20], an intriguing research direction is to study more complicated statistics such as the "higher-degree" ones (e.g., those involving triplets instead of pairs). It would be interesting to see if techniques from linear queries and from our work can be applied to these problems.

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

# A  Additional Preliminaries

We use $d_{\mathrm{tv}}, d_{\mathrm{KL}}$ to denote the total variation (TV) distance and the Kullback–Leibler (KL) divergence between two distributions. Pinsker's inequality states that

$$d_{\mathrm{tv}}(P, Q) \leq \sqrt{\frac{1}{2} \cdot d_{\mathrm{KL}}(P \parallel Q)}, \tag{1}$$

for all distributions $P, Q$.

### Interactive Local DP

For interactive local DP, we consider protocols that proceed in rounds. In each round, the analyst may send a message (which may depend on what the analyst has received in the previous rounds) to each user, who then replies back with a (randomized) response. The DP guarantee is then enforced on the view of the analyst.

### Clipping and Resulting Error

*Clipping* is a standard technique in DP to achieve bounded sensitivity (cf. [ACG+16]). In our interactive algorithm, we will use clipping on real values to bound their sensitivity. More specifically, for $\tau \geq 0$, let $\mathrm{clip}_\tau : \mathbb{R} \to \mathbb{R}$ denote the function:

$$\mathrm{clip}_\tau(x) = \begin{cases} \tau & \text{if } x > \tau, \\ x & \text{if } \tau \geq x \geq -\tau, \\ -\tau & \text{if } -\tau > x. \end{cases}$$

There could be additional errors resulting from clipping. For the purpose of bounding such terms, we will use the following simple lemma:

**Lemma 18.** *Let $X$ denote any random variable over $\mathbb{R}$ and any $\tau \geq 0$. Then, we have*

$$\mathbb{E}[(X - \mathrm{clip}_\tau(X))^2] \leq \sqrt{\mathbb{E}[X^4] \cdot \Pr[|X| > \tau]}.$$

*Proof.* We can bound the LHS term by

$$\begin{aligned}
\mathbb{E}[(X - \mathrm{clip}_\tau(X))^2] &\leq \mathbb{E}[X^2 \cdot \mathbf{1}[|X| > \tau]] \\
&\leq \sqrt{\mathbb{E}[X^4] \cdot \mathbb{E}[\mathbf{1}[|X| > \tau]^2]} \\
&= \sqrt{\mathbb{E}[X^4] \cdot \Pr[|X| > \tau]},
\end{aligned}$$

where in the second step we used the Cauchy–Schwarz inequality. $\qquad\square$

### Factorization vs Approximate-Factorization: Proof of Lemma 10

*Proof of Lemma 10.* Let $W$ be any matrix. By definition of $\zeta$, there exists $\tilde{W}$ such that $\zeta(W; n) = \gamma_2(\tilde{W}) + \|\tilde{W} - W\|_\infty \cdot (\varepsilon\sqrt{n})$. The new protocol $\mathbb{A}'$ simply runs $\mathbb{A}$ but on $\tilde{W}$. Let $\hat{z}$ denote the output of $\mathbb{A}'$. Its MSE can be bounded by

$$\begin{aligned}
\mathbb{E}\left[\left(\hat{z} - h_{\mathbf{x}}^T W h_{\mathbf{x}}\right)^2\right] &= \mathbb{E}\left[\left(\left(\hat{z} - h_{\mathbf{x}}^T \tilde{W} h_{\mathbf{x}}\right) + \left(h_{\mathbf{x}}^T \tilde{W} h_{\mathbf{x}} - h_{\mathbf{x}}^T W h_{\mathbf{x}}\right)\right)^2\right] \\
&\lesssim \mathbb{E}\left[\left(\hat{z} - h_{\mathbf{x}}^T \tilde{W} h_{\mathbf{x}}\right)^2\right] + \mathbb{E}\left[\left(h_{\mathbf{x}}^T \tilde{W} h_{\mathbf{x}} - h_{\mathbf{x}}^T W h_{\mathbf{x}}\right)^2\right] \\
&\lesssim \left(c(n, \varepsilon, k) \cdot \frac{\gamma_2(\tilde{W})^2}{\varepsilon^2 n}\right) + \|\tilde{W} - W\|_\infty^2 \\
&\lesssim c(n, \varepsilon, k) \cdot \frac{\zeta(W; n)^2}{\varepsilon^2 n} + \frac{\zeta(W; n)^2}{\varepsilon^2 n} \\
&\lesssim c(n, \varepsilon, k) \cdot \frac{\zeta(W; n)^2}{\varepsilon^2 n}. \qquad\square
\end{aligned}$$

**Projection Mechanism**

As outlined in Section 1.2, we will use the *projection mechanism* as a subroutine for our algorithm. For $W \in \mathbb{R}^{\ell \times k}$, we define $W^\Delta$ to denote the set $\{Wx \mid x \in \mathbb{R}^k, \|x\|_1 \le 1\}$. The guarantees of the algorithm are summarized below.

**Theorem 19** ([BBNS19]). *For any workload $W \in \mathbb{R}^{\ell \times k}$, there is a non-interactive $\varepsilon$-local DP mechanism $\mathcal{M}$ such that*

$$\mathbb{E}_{\mu \sim \mathcal{M}(\mathbf{x})}[\|\mu - Wh_{\mathbf{x}}\|_2^2] \lesssim \frac{\|W\|_{1\to 2}^2 \sqrt{\log k}}{\varepsilon \sqrt{n}}.$$

*Moreover, the output $\mathcal{M}(W)$ always belongs to $W^\Delta$.*

We provide the proof below for completeness.

*Proof of Theorem 19.* Let $C := \|W\|_{1\to 2}$. The algorithm works as follows:

 ▷ Each user $i \in [n]$ sends $y_i := \mathrm{VRand}_{\varepsilon, C}(W\mathbf{1}_{x_i})$ to the analyst (where $\mathrm{VRand}.(\cdot)$ is from Theorem 9).

 ▷ The analyzer computes $Y := \frac{1}{n} \sum_{i \in [n]} y_i$ and then outputs $\mu := \mathrm{argmin}_{u \in W^\Delta} \|u - Y\|_2$.

The privacy guarantee of the algorithm follows from Theorem 9.

Let $z_i = y_i - W\mathbf{1}_{x_i}$. From Theorem 9, $z_i$ is zero-mean and $\sigma$-sub-Gaussian for $\sigma = O(C/\varepsilon)$. Let $Z := \frac{1}{n} \sum_{i \in [n]} z_i$; we then have that it is zero-mean and $\sigma'$-sub-Gaussian for $\sigma' = \sigma/\sqrt{n}$. By the definition of sub-Gaussian random variable and a union bound, for every $\beta > 0$, with probability $1 - \beta$ the following holds:

$$|\langle W_j, Z\rangle| \lesssim \frac{C^2}{\varepsilon\sqrt{n}} \cdot \sqrt{\log(k/\beta)} \qquad\qquad \forall j \in [k], \qquad\qquad (2)$$

where $W_j$ denote the $j$th column of $W$.

It is well known[7] that if we define $\mu$ as we did (i.e., as projection of $Y$ on $W^\Delta$), then we have

$$\langle \mu - w, \mu - Y\rangle \le 0,$$

for all $w \in W^\Delta$.

Plugging in $w = Wh_{\mathbf{x}}$, we have

$$\begin{aligned}
\|\mu - Wh_{\mathbf{x}}\|_2^2 &= \langle \mu - Wh_{\mathbf{x}}, \mu - Y\rangle + \langle \mu - Wh_{\mathbf{x}}, Z\rangle \\
&\le \langle \mu - Wh_{\mathbf{x}}, Z\rangle \\
&\le \max_{w' \in W^\Delta} \langle w', Z\rangle \\
&= \max_{j \in [k]} |\langle W_j, Z\rangle|,
\end{aligned}$$

where the equality follows from the fact $W^\Delta$ is the convex hull of $W_1, \ldots, W_k, -W_1, \ldots, -W_k$.

Putting together (2) and the above then yields

$$\mathbb{E}[\|\mu - Wh_{\mathbf{x}}\|_2^2] \lesssim \frac{C^2\sqrt{\log k}}{\varepsilon\sqrt{n}}. \qquad\qquad \square$$

# B  On the Lower Bound for Linear Queries from [ENU20]

In this section, we provide details on how we can interpret the bounds of [ENU20] as stated in the form of Theorem 4. The lower bound in [ENU20] is originally for the $\ell_\infty$-error; we make the observation below that a simple modification of their proof can be made so that it applies to mMSE. Note that, in addition to getting a stronger result in terms of error metric (because a lower bound on

---

[7]See e.g., [Bub15, Lemma 3.1].

mMSE implies the same lower bound on $\ell_\infty$-error), we also get a quantitatively stronger bound that does not depend on $k$ because we avoid having to take a union bound over $k$ queries (which was required for the proof for $\ell_\infty$-error in [ENU20]).

In [ENU20, Section 3.4], it was shown that w.l.o.g. it suffices to consider "symmetric" workload matrix $W$ (i.e., ones that can be written as $[W', -W']$ for some $W'$). We will thus do so throughout the rest of this section. Following their notation, we also assume that $W \in \mathbb{R}^{k \times \mathcal{X}}$, i.e., $\mathcal{X}$ is the input space and there are $k$ linear queries. We also recall the following two lemmas from their paper.

**Lemma 20** ([ENU20, Lemma 11]). *Let $\varepsilon \in (0, 1]$. For any distribution $\lambda_1, \ldots, \lambda_m, \mu_1, \ldots, \mu_m$ on $\mathcal{X}$, distribution $\pi$ over $[m]$ and $\varepsilon$-local DP randomizer $\mathcal{R}$, we have*[8]

$$\mathbb{E}_{v \sim \pi}[d_{\mathrm{KL}}(\mathcal{R}(\lambda_v)^n \| \mathcal{R}(\mu_v)^n)] \lesssim n\varepsilon^2 \cdot \|M\|^2_{\ell_\infty \to L_2(\pi)},$$

*where $M \in \mathbb{R}^{m \times \mathcal{X}}$ is a matrix such that $M_{v,x} = \lambda_v(x) - \mu_v(x)$.*

We note that in [ENU20, Lemma 11], the LHS is not exactly the same as in our version above. However, inspecting the very first inequality from their proof shows that their bound goes through the quantity on the LHS of Lemma 20.

The next lemma, which describes the properties of the hard distributions, is arguably the main technical contribution of the lower bound in [ENU20].

**Lemma 21** ([ENU20, Lemma 21]). *Let $W \in \mathbb{R}^{k \times \mathcal{X}}$ be any symmetric workload matrix and $\alpha > 0$. There exist*[9] *$\xi \geq 0$, and distributions $\tilde{\lambda}_1, \ldots, \tilde{\lambda}_k, \tilde{\mu}_1, \ldots, \tilde{\mu}_k$ on $\mathcal{X}$ and $\tilde{\pi}$ over $[k]$ such that*

> *(i) $(W\tilde{\lambda}_v)_i = 0$ for all $i, v \in [k]$.*
>
> *(ii) for all $v \in \mathrm{supp}(\tilde{\pi})$, $(W\tilde{\mu}_v)_v \gtrsim \frac{\xi + \alpha}{\log(\|W\|_\infty / \alpha)}$.*[10]
>
> *(iii) Let $\tilde{M}$ be defined similarly as in Lemma 20. Then $\|\tilde{M}\|_{\ell_\infty \to L_2(\tilde{\pi})} \lesssim \frac{\xi}{\gamma_2(W, \alpha)}$.*

It will in fact be easier to work with the following version of the lemma, which is implicit in the proof of [ENU20, Theorem 22]:

**Lemma 22.** *Let $W \in \mathbb{R}^{k \times \mathcal{X}}$ be any symmetric workload matrix and $\alpha > 0$. There exist distributions $\tilde{\lambda}_1, \ldots, \tilde{\lambda}_k, \tilde{\mu}_1, \ldots, \tilde{\mu}_k$ on $\mathcal{X}$ and $\tilde{\pi}$ over $[k]$ such that*

> *(i) $(W\tilde{\lambda}_v)_i = 0$ for all $i, v \in [k]$.*
>
> *(ii) for all $v \in \mathrm{supp}(\tilde{\pi})$, $(W\tilde{\mu}_v)_v \gtrsim \frac{\alpha}{\log(\|W\|_\infty / \alpha)}$.*
>
> *(iii) Let $\tilde{M}$ be defined similarly as in Lemma 20. Then $\|\tilde{M}\|_{\ell_\infty \to L_2(\tilde{\pi})} \lesssim \frac{\alpha}{\gamma_2(W, \alpha)}$.*

*Proof Sketch.* This is constructed by taking $\tilde{\lambda}_1, \ldots, \tilde{\lambda}_k, \tilde{\mu}_1, \ldots, \tilde{\mu}_k$ and $\tilde{\pi}$ from Lemma 21. Then, replace each $\tilde{\mu}_v$ by the mixture $(1 - \beta) \cdot \tilde{\lambda}_v + \beta \cdot \tilde{\mu}_v$ where $\beta = \min\{1, \alpha/\xi\}$, for all $v \in [k]$. Both the lower bound for $(W\tilde{\mu}_v)_v$ and the upper bound for $\|\tilde{M}\|_{\ell_\infty \to L_2(\tilde{\pi})}$ scale linearly with $\beta$, yielding the desired bounds. $\square$

We can now prove the following, which is a more qualitative version of the lower bound in Theorem 4.

**Theorem 23.** *Let $W \in \mathbb{R}^{k \times \mathcal{X}}$ be any symmetric workload matrix. Any non-interactive $\varepsilon$-local DP mechanism must incur an* mMSE *at least* $\Omega \left( \frac{\zeta(W,n)^2}{\varepsilon^2 n} \cdot \frac{1}{\log\left( \frac{\varepsilon^2 n \|W\|^2_\infty}{\zeta(W,n)^2} \right)^2} \right)$.

---

[8]Here $\|M\|^2_{\ell_\infty \to L_2(\pi)} := \max_{\|x\|_\infty = 1} \|Mx\|_{L_2(\pi)}$ where $\|a\|_{L_2(\pi)} := \sqrt{\sum_{v \in [m]} \pi(v)a_v^2}$. Note that we will not be dealing with this quantity further than here.

[9]In [ENU20], $\xi$ is related to the dual solution as $\xi = W \bullet U - \alpha$ where $U$ is the dual witness of the approximate factorization norm, i.e., one with $\gamma_2(W) = \frac{W \bullet U - \alpha \|U\|_1}{\gamma_2^*(U)}$. (See [ENU20, Section 2.3] for more details.) However, these specifics are not used in the remainder of the proof.

[10]Note that the $\|W\|_\infty$ term does not show up in [ENU20] since they assume that $\|W\|_\infty \leq 1$.

*Proof.* For notational convenience, we write $\zeta$ as a shorthand for $\zeta(W, n)$. Let $\alpha = \frac{C_1\zeta}{\varepsilon\sqrt{n}}$ where $C_1 \in (0, 0.1)$ is a sufficiently small constant. Suppose for the sake of contradiction that there exists a non-interactive $\varepsilon$-local DP protocol (whose randomizer is $\mathcal{R}$) with mMSE at most $(\alpha')^2$ for $\alpha' = \frac{C_2\alpha}{\log(\|W\|_\infty/\alpha)}$ where $C_2 \in (0, 0.1)$ is a sufficiently small constant.

Recall the definition of $\zeta$; we must have $\zeta \leq \gamma_2\left(W, \frac{\zeta}{2\varepsilon\sqrt{n}}\right) + \frac{\zeta}{2}$. This gives

$$\gamma_2\left(W, \frac{\zeta}{2\varepsilon\sqrt{n}}\right) \geq \frac{\zeta}{2}.$$

Since $\gamma_2(W, \cdot)$ is an increasing function, we thus have

$$\frac{\alpha\varepsilon\sqrt{n}}{\gamma_2(W, \alpha)} \leq C_1. \tag{3}$$

Let $\tilde{\lambda}_1, \ldots, \tilde{\lambda}_k, \tilde{\mu}_1, \ldots, \tilde{\mu}_k$ and $\tilde{\pi}$ be as in Lemma 22. By Lemma 20 and Lemma 22(iii), we have

$$\mathbb{E}_{v\sim\pi}[d_{\mathrm{KL}}(\mathcal{R}(\lambda_v)^n \parallel \mathcal{R}(\mu_v)^n)] \lesssim \left(\frac{\alpha\varepsilon\sqrt{n}}{\gamma_2(W, \alpha)}\right)^2. \tag{4}$$

On the other hand, since the protocol has mMSE at most $(\alpha')^2$, we may use the following algorithm to distinguish $\mathcal{R}(\lambda_v)^n$ from $\mathcal{R}(\mu_v)^n$: let the analyst computes an estimate for $Wh_{\mathbf{x}}$ (using the $n$ samples from $\mathcal{R}(\cdot)$ provided). If the estimate has absolute value less than $10\alpha'$, then output $\lambda_v$; otherwise, output $\mu_v$. From Lemma 22(i)(ii), it is not hard to see that, for $C_2$ that is sufficiently small, this algorithm is correct with probability at least 2/3. This means that $d_{\mathrm{tv}}(\mathcal{R}(\lambda_v)^n, \mathcal{R}(\mu_v)^n) \geq 1/3$. Pinsker's inequality (i.e., eq. (1)) then yields $d_{\mathrm{KL}}(\mathcal{R}(\lambda_v)^n \parallel \mathcal{R}(\mu_v)^n) \gtrsim 1$. Comparing this with the above eq. (4), we get $\frac{\alpha\varepsilon\sqrt{n}}{\gamma_2(W, \alpha)} \gtrsim 1$, which contradicts eq. (3) when $C_1$ is sufficiently small. $\square$

## C  Three-Round Algorithm

In this section, we describe and analyze our interactive algorithm. The error guarantees of the algorithm are stated below. (Note that Theorem 24 is a more precise version of Theorem 8.)

**Theorem 24.** *For any $W \in \mathbb{R}^{k\times k}$ and $n \geq \tilde{\Omega}\left(\frac{\gamma_2(W)^4 \log k}{\|W\|_\infty^4 \varepsilon^2}\right)$, there is a three-round $\varepsilon$-DP algorithm for estimating quadratic forms on $W$ such that the MSE is $O\left(\frac{\|W\|_\infty^2}{\varepsilon^2 n}\right)$.*

*Proof.* Throughout the proof, we assume that $n \geq Q \cdot \frac{\gamma_2(W)^4 \log k}{\|W\|_\infty^4 \varepsilon^2} \cdot \log\left(\frac{10\gamma_2(W)\log k}{\|W\|_\infty \varepsilon}\right)$, where $Q$ is a sufficiently large constant. Recall from the definition of $\gamma_2(W)$ that there must exist $L, R$ such that $W = L^T R$ and $\|L\|_{1\to 2} = \|R\|_{1\to 2} = \sqrt{\gamma_2(W)} =: C$.

**Algorithm Description.** Let $\bar{\varepsilon} = \varepsilon/4$ and $\tau = 4\|W\|_\infty$. The algorithm works as follows:

▷ **First Round:**  Run the $\bar{\varepsilon}$-local DP protocol from Theorem 19 to get an estimate $\mu^R$ of $Rh_{\mathbf{x}}$.

▷ **Second Round:**

    ▷ The analyzer forwards $\mu^R$ to all users.

    ▷ Each user $i \in [n]$ sends the following to the analyst:

        ▷ $y_i^L \leftarrow \mathrm{VRand}_{\bar{\varepsilon}, C}(L\mathbf{1}_{x_i})$ (where VRand$.(\cdot)$ is from Theorem 9).

        ▷ $a_i \leftarrow \langle L\mathbf{1}_{x_i}, \mu^R\rangle + \kappa_i$ where $\kappa_i \sim \mathrm{Lap}(2\|W\|_\infty/\bar{\varepsilon})$ back to the analyst.

▷ **Third Round:**

    ▷ The analyst then computes $Y^L := \frac{1}{n}\sum_{i\in[n]} y_i^L$ and forwards it to the users.

    ▷ Each user $i \in [n]$ sends $v_i \leftarrow \mathrm{clip}_\tau(\langle Y^L, R\mathbf{1}_{x_i} - \mu^R\rangle) + z_i$ where $z_i \sim \mathrm{Lap}(2\tau/\bar{\varepsilon})$.

Finally, the analyst outputs $\frac{1}{n}\left(\sum_{i\in[n]} a_i\right) + \frac{1}{n}\left(\sum_{i\in[n]} v_i\right)$.

**Privacy Analysis.** Each user $i$'s input is used four times:

▷ To produce $\mu^R$. This step is $\bar{\varepsilon}$-DP by Theorem 19.

▷ To produce $y_i^L$. This step is $\bar{\varepsilon}$-DP by Theorem 9.

▷ To produce $a_i$. From the guarantee of Theorem 19, $\mu_R$ belong to $R^\Delta$. As a result, $|\langle L\mathbf{1}_{x_i}, \mu_R \rangle| \leq \|W\|_\infty$. In other words, the sensitivity of $\langle L\mathbf{1}_{x_i}, \mu_R \rangle$ (as a function of $x_i$) is at most $2\|W\|_\infty$. Since $\kappa_i$ is sampled from $\mathrm{Lap}(2\|W\|_\infty/\bar{\varepsilon})$, this step is also $\bar{\varepsilon}$-DP.

▷ To produce $v_i$. Due to clipping, the sensitivity of $\mathrm{clip}_\tau(\langle Y^L, R\mathbf{1}_{x_i} - \mu_R \rangle)$ (as a function of $x_i$) is at most $2\tau$. Thus, since we are adding a noise $z_i$ drawn from $\mathrm{Lap}(2\tau/\bar{\varepsilon})$, this step is also $\bar{\varepsilon}$-DP.

Hence, by the basic composition theorem, the entire algorithm is $\varepsilon$-local DP as desired.

**Utility Analysis.** First, notice that, for a fixed $\mu_R \in R^\Delta$, we have
$$
h_\mathbf{x}^T W h_\mathbf{x} = \langle Lh_\mathbf{x}, Rh_\mathbf{x} \rangle = \langle Lh_\mathbf{x}, \mu_R \rangle + \langle Lh_\mathbf{x}, Rh_\mathbf{x} - \mu_R \rangle
$$
$$
= \frac{1}{n} \sum_{i \in [n]} \langle L\mathbf{1}_{x_i}, \mu_R \rangle + \frac{1}{n} \sum_{i \in [n]} \langle Lh_\mathbf{x}, R\mathbf{1}_{x_i} - \mu_R \rangle.
$$

Let $z_i^L = y_i^L - L\mathbf{1}_{x_i}$. From Theorem 9, $z_i^L$ is zero-mean and $\sigma$-sub-Gaussian for $\sigma = O(C/\varepsilon)$. Let $Z^L := \frac{1}{n} \sum_{i \in [n]} z_i^L$; we then have that $Z^L$ is zero-mean and $\sigma'$-sub-Gaussian for $\sigma' = \sigma/\sqrt{n}$. Note that $Y^L = Lh_\mathbf{x} + Z^L$. Furthermore, let us write $\theta_i$ as a shorthand for $\mathrm{clip}_\tau(\langle Y^L, R\mathbf{1}_{x_i} - \mu_R \rangle) - \langle Y^L, R\mathbf{1}_{x_i} - \mu_R \rangle$.

The MSE can be bounded by

$$
\mathbb{E}\left[\left(\frac{1}{n}\left(\sum_{i \in [n]} a_i\right) + \frac{1}{n}\left(\sum_{i \in [n]} v_i\right) - h_\mathbf{x}^T W h_\mathbf{x}\right)^2\right]
$$
$$
= \mathbb{E}\left[\left(\frac{1}{n}\sum_{i \in [n]} \kappa_i + \frac{1}{n}\sum_{i \in [n]} z_i + \langle Z^L, Rh_\mathbf{x} - \mu_R \rangle + \frac{1}{n}\sum_{i \in [n]} \theta_i\right)^2\right]
$$
$$
\lesssim \mathbb{E}\left[\left(\frac{1}{n}\sum_{i \in [n]} \kappa_i\right)^2\right] + \mathbb{E}\left[\left(\frac{1}{n}\sum_{i \in [n]} z_i\right)^2\right] + \mathbb{E}\left\langle Z^L, Rh_\mathbf{x} - \mu_R \right\rangle^2 + \mathbb{E}\left[\left(\frac{1}{n}\sum_{i \in [n]} \theta_i\right)^2\right]
$$
$$
\lesssim \frac{\|W\|_\infty^2}{\varepsilon^2 n} + \mathbb{E}\left\langle Z^L, Rh_\mathbf{x} - \mu_R \right\rangle^2 + \frac{1}{n}\sum_{i \in [n]} \mathbb{E}\left[\theta_i^2\right], \tag{5}
$$

where the last step is by applying the Cauchy–Schwarz inequality to the last term.

To handle the middle term in eq. (5), note that $Z^L$ is independent of $\mu_R$ and, as stated earlier, is $\sigma'$-sub-Gaussian. Thus, we have
$$
\mathbb{E}\left\langle Z^L, Rh_\mathbf{x} - \mu_R \right\rangle^2 = \mathbb{E}_{\mu_R} \mathbb{E}_{Z^L} \left\langle Z^L, Rh_\mathbf{x} - \mu_R \right\rangle^2
$$
$$
\lesssim \mathbb{E}_{\mu_R} (\sigma')^2 \|Rh_\mathbf{x} - \mu_R\|_2^2
$$
$$
\text{(From Theorem 19)} \lesssim (\sigma')^2 \cdot \frac{C^2 \sqrt{\log k}}{\varepsilon \sqrt{n}}
$$
$$
\lesssim \frac{C^4 \sqrt{\log k}}{\varepsilon^3 n^{3/2}}
$$
$$
\lesssim \frac{\|W\|_\infty^2}{\varepsilon^2 n}, \tag{6}
$$

where the last inequality is due to our assumption on $n$.

As for the last term in eq. (5), notice that
$$
\langle Y^L, R\mathbf{1}_{x_i} - \mu_R \rangle = \langle Z^L, R\mathbf{1}_{x_i} - \mu_R \rangle + \langle Lh_\mathbf{x}, R\mathbf{1}_{x_i} - \mu_R \rangle.
$$

Since $\mu_R$ belongs to $R^\Delta$, we have $\frac{1}{2}\left(R\mathbf{1}_{x_i} - \mu_R\right) \in R^\Delta$, which implies that $|\langle Lh_\mathbf{x}, R\mathbf{1}_{x_i} - \mu_R \rangle| \leq 2\|W\|_\infty$ and $\|R\mathbf{1}_{x_i} - \mu_R\|_2 \leq 2\tilde{C}$. As such, we have

$$
\begin{aligned}
\mathbb{E}\left\langle Y^L, R\mathbf{1}_{x_i} - \mu_R \right\rangle^4 &\lesssim \mathbb{E}\left\langle Z^L, R\mathbf{1}_{x_i} - \mu_R \right\rangle^4 + \|W\|_\infty^4 \\
&\lesssim (\sigma')^4 \cdot C^4 + \|W\|_\infty^4 \\
&\lesssim \frac{C^8}{\varepsilon^4 n^2} + \|W\|_\infty^4 \\
&\lesssim \|W\|_\infty^4,
\end{aligned}
$$

where the last inequality is due to our assumption on $n$ and from $\gamma_2(W) \geq \|W\|_\infty$.

Meanwhile, since $\tau = 4\|W\|_\infty$, we have

$$
\begin{aligned}
\Pr[|\left\langle Y^L, R\mathbf{1}_{x_i} - \mu_R \right\rangle| > \tau] &\leq \Pr[|\left\langle Z^L, R\mathbf{1}_{x_i} - \mu_R \right\rangle| > \tau/2] \\
&\leq \exp\left(-\Omega(\tau/(\sigma' \cdot 2C))^2\right) \\
&= \exp\left(-\Omega(\varepsilon\|W\|_\infty \sqrt{n}/C^2)^2\right) \\
&\leq \frac{1}{\varepsilon^4 n^2},
\end{aligned}
$$

where the last inequality is again due to our assumption on $n$ and from $\gamma_2(W) \geq \|W\|_\infty$.

Thus, we may apply Lemma 18 to conclude that

$$
\mathbb{E}[\theta_i^2] \lesssim \sqrt{\|W\|_\infty^4 \cdot \frac{1}{\varepsilon^4 n^2}} = \frac{\|W\|_\infty^2}{\varepsilon^2 n}. \tag{7}
$$

Combining eqs. (5) to (7), the MSE of the estimate is at most $O\left(\frac{\|W\|_\infty^2}{\varepsilon^2 n}\right)$ as desired. $\qquad\square$

## C.1 On Separating Non-Interactive and Interactive Local DP

We end by observing that our interactive local DP algorithm (Theorem 24) together with the non-interactive lower bound (Theorem 6 and particularly, the more quantitative version, Theorem 23) gives an asymptotic separation on the MSE achievable by $\varepsilon$-local DP interactive algorithms and those that are non-interactive, as long as we pick $W$ together with $n \in \mathbb{N}$ such that the following holds:

▷ $n \gtrsim \frac{\gamma_2(W)^4 \log k}{\|W\|_\infty^4} \cdot \log\left(\frac{10\gamma_2(W) \log k}{\|W\|_\infty}\right)$ and

▷ $\frac{\zeta(W;n)}{\|W\|_\infty} \geq \omega(\log n)$,

where the first condition is from Theorem 24 (and letting $\varepsilon = 1$) and the second condition ensures that the error from Theorem 24 is asymptotically larger than the lower bound from Theorem 23.

It is simple to verify that it suffices to pick $W$ such that $\|W\|_\infty = 1, \gamma_2(W, 0.1), \gamma_2(W) = \Theta(k^c)$ for any constant $c > 0$ and pick $n$ to precisely satisfy the bound in the first condition. Again, it is not hard to construct such a matrix e.g., by taking $W$ to be a random $(k \times k)$ matrix where each entry is an i.i.d. Rademacher random variable, which yields $\gamma_2(W, 0.1), \gamma_2(W) = \Theta(\sqrt{k})$ w.h.p. [LMSS07].

