# OpenReview forum: "On Computing Pairwise Statistics with Local Differential Privacy"
_NeurIPS.cc/2023/Conference — NeurIPS 2023 poster_

### Official Review · Reviewer_YTda · 2023-07-06

**Soundness:** 3 good
**Presentation:** 2 fair
**Contribution:** 3 good
**Rating:** 5
**Confidence:** 3

**Summary:**

In this paper, the authors analyzed the problem of privately computing the quadratic form in the model of differential privacy, and provide non-interactive local DP algorithm with MSE upper/lower bounds with gap log(k). The paper further develops results for an interactive algorithm for the same problem and proves analogous bounds.

**Strengths:**

The bound results for both non-interactive and interactive local DP algorithms are generic. The authors provided a complete analysis on the non-interactive algorithm along with its bound results.

**Weaknesses:**

Compared to the linear queries result, the MSE bounds of the non-interactive local DP mechanism for estimating quadratic forms reveal a noticeable gap depending on $k$, the dimension of matrix W.

**Questions:**

1. In addition to the discussions in the paper, can the author provide some intuitions on if the upper bound of the non-interactive local DP mechanism for estimating quadratic forms should depend on k?

2. Can the author provide some intuition on why the definition of $\gamma_2(W,\alpha)$ uses infinity norm instead of other norms? Is there any complexity results on other norms?

**Limitations:**

Compared to the similar sample complexity results for computing the statistical queries in the models of differential privacy, the sample complexity results for computing the pairwise statistics in local DP reveals a gap that depends on the dimension of the matrix $W$.

---

> ### Author Rebuttal · Authors · 2023-08-08
>
> Thank you for your review & questions. Please find the answers to your questions below.
>
> 1. This is indeed a great question and is why we also pose it as an open question in our paper (lines 344-345). We conjecture this might be necessary since even the seemingly simpler heavy-hitter problem also requires a dependency on $k$ (e.g,. [Bun et al., PODS’10]).  However, this is just an intuition since we are not aware of a formal reduction between computing heavy-hitters and computing pairwise statistics.
>
>     Furthermore, note that the original work of [ENU20] on linear queries *also contained a $\\log k$ gap*. (In particular, $\\gamma_2(W, \\alpha)$ is required to be at least $\\log k$ for the result to hold; see Theorem 22 in their arxiv version.) It is only in our paper that this gap is closed, due to a simple tweak of the proof; see Appendix B for more details. We will make sure to emphasize this point more clearly in the revision.
>
>
> 2. Roughly speaking, the infinity norm is used due to the fact that we often want a worst case guarantee across all linear queries (i.e., $mMSE$ error). Recall that in the matrix mechanism we output $L(R h_x + z)$ where $z$ is the noise term. Even if the noise term $z$ is zero, our output is still $L R h_x$. Now, it turns out that we can find $h_x$ and $j$ such that the difference between $(LR h\_x)\_j$ (our answer for the $j$-th query) and $(W h\_x)\_j$ (the correct answer for the $j$-th query) is proportional to $\\|LR - W\\|\_{\\infty}$. (Namely, if $(LR - W)\_{ij}$ is the largest entry in absolute value, then we just let $x\_1 = \dots = x\_n = i$.) Since our reduction uses the $mMSE$ error measure, we consider the definition of $\\gamma_2(W, \\alpha)$ as stated in the paper.
>
>     If we want some average guarantee, then it seems plausible to change the infinity norm to, e.g., the Frobenius norm. However, we are not aware of previous results on this.

---

> > ### Comment · Reviewer_YTda · 2023-08-14
> >
> > Thank you for your response, and I will keep my score.

---

### Official Review · Reviewer_xtaj · 2023-07-06

**Soundness:** 2 fair
**Presentation:** 2 fair
**Contribution:** 2 fair
**Rating:** 3
**Confidence:** 3

**Summary:**

This paper studies the computing of pairwise statistics with local differential privacy by considering the quadratic form computation.  In order to obtain the lower boudn and uppoer bounds, it proposes the inter-reductions between quadratic forms and linear queries.

**Strengths:**

Studying pair statistics with local differential privacy is intersting.

**Weaknesses:**

(1)  The presentation needs some improvement.   There are many grammmatical mistakes in many  places.  The preliminary part is quite messy: a formal definition of central differetial privacy is given but the definition of local DP is informal.   Randomized response mechanisms are not defined in the paper.
(2)  I don't see the significance of the two reductions in Section 1.2.    Either the reductions are trivial or the subtlety has not been made explicit yet.


**Questions:**

Q1:  Could you pls explain the logic behind the sentence in Line 150?

Q2: Could you explain more the matrix mechanism especially the factorization? (Line 159)

Q3:  I don't see the reasoning in Line 288 especially the second line there about "cross terms".

**Limitations:**

This paper highly depends on the results in ENU20. I don't see any significant technical contribution in this paper

---

> ### Author Rebuttal · Authors · 2023-08-08
>
> ## High-level Response:
>
> We completely agree with the reviewer that the reductions in Section 1.2 are simple, _but only in hindsight!_ On the other hand, we think of this as a significant contribution of our work for the following reason.  While previous works [BBGK20 (AISTATS’20), CM22 (VLDB’23), BHBFG+22 (NeurIPS’22)] had elaborate and specific algorithms for _each_ specific kernel $f$, our reduction gives algorithms for *all* kernels that essentially match (and sometimes even improve upon) the guarantees in previous works. Similarly, none of the previous work showed any lower bounds for their problem, meanwhile our reduction gives nearly tight lower bounds for all possible kernels in the non-interactive setting (Theorem 6). In summary, our reductions give simple, tight, and extremely general results for the problem of computing pairwise statistics.
>
>
> Regarding the comparison to [ENU20], we wish to point out that we study a *completely different problem* (computing pairwise statistics) compared to that paper (computing linear queries). Indeed, one of our main (and arguably surprising) contributions is to show that these two ostensibly unrelated problems are intimately related.
>
>
> We hope the reviewer will consider these points in their (re)evaluation of the paper.
>
>
> ## Q1:
>
>
> First, we apologize for the typos: on line 145 (and 131) $( \\hat{z}_1, \\dots, \\hat{z}_j )$ should be changed to $( \\hat{z}_1, \\dots, \\hat{z}_k )$.
>
>
> Now, on to line 150: If we do not add noise at all, then we would set $\\kappa\_j = 0$ and the $( {\\hat{z}}\_1, {\\dots}, {\\hat{z}}\_k )$ on line 145 are exactly equal to the answer of the linear queries.
> This means that $o_j = \\hat{z}\_{x\_j} = {{(W h\_x)}\_{x\_j}} + 0 = {{1_{x_j}}^T} W h_x$.
> Averaging this over $j = 1, …, n$, we get $\\frac{1}{n}(o_1 + \\dots o_n) = \\frac{1}{n}\\left(1_{x_1}^T + \\cdots + 1_{x_n}^T\\right) W h_x = h_x^T W h_x$, whereas the last inequality simply follows from the definition of normalized histogram $h_x$ (line 33).
>
>
> ## Q2:
>
>
> Matrix mechanism is a standard tool in DP that dates back to (at least) the paper “Optimizing linear counting queries under differential privacy” from Li et al. in PODS'10. Given the long history of the topic, it is impossible to cover it thoroughly below; [ENU20] and references therein contains a more detailed picture.
>
>
> Nevertheless, we will try to explain it here. The matrix mechanism is based on the following idea.  If the linear queries in $W$ are similar (or correlated in certain ways), it is not optimal to add independent noise to them. For example, let’s say that all linear queries in $W$ are identical. Namely, all the rows of $W$ are the same. The “trivial” algorithm here is to add independent noise to each answer. Due to (the advanced) composition of DP, this will mean that we will have to scale the noise by a factor of roughly $\\sqrt{k}$. Meanwhile, a much better algorithm is to compute the answer of just a single query, add noise to it, and then use this noisy value as the answer to all queries. This has the same noise as if we were to answer a single query, so we get a lot of savings by doing so! Here we can think of it as factoring $W = L \\cdot R$ where $L$ is the $(k \\times 1)$ all-one matrix and $R$ as the $(1 \\times k)$ matrix containing a single row of $W$. The aforementioned algorithm is thus to compute $y = R h_x + z$ where $z$ is the noise and answer $L y$. The matrix mechanism is the generalization of this, which allows arbitrary factorizations $L, R$. It turns out that $\\gamma_2(W)$ is exactly the error of the matrix mechanism (after some scaling) and that the optimal factorization can be done efficiently.
>
>
> ## Q3:
>
>
> We are simply using the identity $(a + b)^2 \\leq 2a^2 + 2b^2$ here, so there is no cross term to be taken care of.

---

### Official Review · Reviewer_6AxR · 2023-07-06

**Soundness:** 4 excellent
**Presentation:** 3 good
**Contribution:** 3 good
**Rating:** 6
**Confidence:** 4

**Summary:**

This paper studies the problem of computing Quadratic Forms $h_x^TWh_x$ under Local Differential Privacy, where $h_x$ is the normalized histogram representation of a vector $x\in [k]^n$. In particular, reductions to and from the problem of computing linear queries are established, through which algorithms and tight bound results (in mean squared error) consequently follow from those in linear queries. The reduction from linear queries is built on the observation that the $j$th entry $(Wh_x)_j$ of the linear query can be obtained from three quadratic forms:

($h_{x\cup 1_j}^TWh_{x\cup 1_j}$), ($h_{x}^TWh_{x}$) and ($h_{1_j}^TWh_{1_j}$).

The reduction to linear queries is built on two observations: 1) $h_x^T W h_x^T$ is a linear query on $h_x$ with weights $Wh_x$ (also a linear query); 2) $h_x^T W h_x^T$ is an inner product of $Lh_x$ and $Rh_x$, where $W=L^T R$. The second approach can be further refined, where using a JL projection on $L$ and $R$ allows the magnitudes of the noise terms associated with privatizing $Lh_x$ and $Rh_x$ to be reduced to $O(\log n)$ from $O(k)$.

**Strengths:**

- This paper presents a generic framework for estimating pairwise statistics that can be expressed as quadratic forms (although utility in some statistics can be improved by exploiting their specific properties).
- The ideas used are interesting and natural, the presented analysis is solid.
- They discuss both interactive and non-interactive settings, and show that computing pairwise statistics separate interactive and non-interactive local DP.

**Weaknesses:**

- In the proof of Theorem 5, the condition on $\alpha$ requires $\varepsilon \ge 1/\sqrt{n}$, but $l=O(\log(k) \varepsilon^2 n)$ requires $\varepsilon < 1$ for $l< n$? These place $\varepsilon$ in a restrictive range.
- Although the main paper focuses on presenting reductions in the non-interactive setting, there are places where the discussions of interactive and non-interactive appear together, which caused some distraction for me. I think all of the discussion on the interactive setting can be moved to its own section toward the end.

**Questions:**

- In the proof of Theorem 14, line 286 requires $W$ to be symmetric?

**Limitations:**

N/A.

---

> ### Author Rebuttal · Authors · 2023-08-08
>
> Thank you for your review & questions. Please find the answers to your questions below.
>
> ## $\\epsilon$ value in  Theorem 5:
> We remark that the lower bound requirement $\\epsilon \\geq 1/\\sqrt{n}$ for non-trivial utility is present in essentially _all known_ local DP results and is supported by Theorem 6 (which implies that non-trivial utility is impossible when $\\epsilon \ll \\tilde{O}(1/\sqrt{n})$). In fact, to the best of our knowledge, this is required to get sub-constant error even for the very simple problem of averaging binary values.
>
> On the other hand, there is no upper bound required on $\\epsilon$. Note that our proof of Theorem 5 works for arbitrarily large $\\epsilon$ and the condition $\\ell \\leq O(\\log(k)\\epsilon^2 n)$ does *not* enforce any upper bound on $\\epsilon$.
>
> ## Writing (Interactive vs Non-Interactive):
> Thanks for your suggestion and apologies for causing the distraction.  In the revision, we will consider separating the discussions on the interactive vs non-interactive in a more streamlined way.
>
>
> ## Theorem 14:
> We implicitly assume that $W$ is symmetric, which is the case when we construct $W$ from a kernel $f$ as specified on line 293. Indeed, line 286 uses symmetry of $W$ as you said. We will make sure to clarify this in the revision. (Note that our algorithms work even for asymmetric $W$. However, the lower bound is harder to handle due to the cross terms.)

---

> > ### Comment · Reviewer_6AxR · 2023-08-12
> >
> > Thank you for your response and clarification.

---

### Official Review · Reviewer_q1oD · 2023-07-09

**Soundness:** 4 excellent
**Presentation:** 4 excellent
**Contribution:** 3 good
**Rating:** 8
**Confidence:** 3

**Summary:**

The paper considers the computation of quadratic forms of histograms under local differential privacy (LDP). Previously, special cases of this problem have been studied, but this paper presents a general theory analogous to the existing theory for linear queries. The problem is studied both in the standard LDP setting and using "interactive", multi-round protocols where each user outputs an LDP value in every round. The 9-page submission focuses on the standard, non-interactive setting. Lower and upper bounds are presented that are tight up to polylogarithmic factors. A sketch of interactive protocols (with better utility) is presented, with details in supplementary material.

**Strengths:**

- Identifies a natural problem for which special cases have been studied before and presents a general theory
- The results are tight (up to polylogarithmic factors) for *every* quadratic form
- The techniques for multi-round LDP protocols are particularly interesting, and separate single- and multi-round LDP for a natural problem
- The mechanisms are simple to describe
- The paper is very well-written

**Weaknesses:**

- Though LDP has been deployed in practice, its general usefulness has sometimes been questioned, since it tends to require very large data sizes to get good utility

**Questions:**

- Can you comment on potential practical use of your results? Are there large factors hidden in big-O notation, and in particular what kind of data size is needed to get good utility for the examples in Corollary 7?
- The shuffle model is sometimes used to amplify privacy of LDP protocols. What results in the this model are implied by your results?
- In line 36, I suppose $\mathcal{X}$ should be $\mathcal{X}^2$?

**Limitations:**

Since there are no experiments, I would have like at least some discussion of the extent to which the protocols proposed might be practical, or whether the contribution is considered purely theoretical.

---

> ### Author Rebuttal · Authors · 2023-08-08
>
> Thank you for your review & questions. Please find the answers to your questions below.
>
> - While we omitted the constants (following the precedence of not reporting exact constants in most previous work), it is not hard to compute them in Corollary 7.  We will consider adding them in the revision. In terms of practicality, our algorithms subsume those in previous work such as [BBGK20, BHBFG+22]. Therefore, the practicality is similar, e.g., our protocol for Gini’s diversity index has less than 0.01 error for $\epsilon=3, k \leq 1000$ and $n \geq 100,000$.
>
> - This is a great question. Our non-interactive algorithm only requires a vector-summation primitive (where each vector has norm at most $C$). Since known protocols for vector summation in the Shuffle model achieve an RMSE of $O\\left(C\\cdot\\sqrt{\\log(1/\\delta)}/\\epsilon\\right)$ [Balle et al., CCS 2020], we can simply replace Theorem 9 in our paper by this protocol and reduce MSE by a factor of $n$ in Theorem 5 and Corollary 7. Thank you for suggesting this; we will add this to the revision.
>
> - Re line 36: Yes, $\mathcal{X}$ should be replaced by $\mathcal{X}^2$. Thank you for pointing this out; we will fix this in the revision.

---

> > ### Comment · Reviewer_q1oD · 2023-08-10
> > **Response to rebuttal**
> >
> > Thanks for the additional details. I think it would make sense to add something along those lines to the paper.

---

### Decision · Program_Chairs · 2023-09-21

**Decision:**

Accept (poster)

**Comment:**

This paper investigates the private computation of pairwise statistics on discrete data (a.k.a., quadratic forms). Relying on connections to linear queries, they provide lower bounds as well as general algorithms with guaranteed error bounds. Most reviewers support acceptance, praising the nice theory, the improvements compared to prior work, and the interesting connections with other private estimation problems. One reviewer voted for rejection mostly based on presentation issues, which in my opinion are rather minor and were clarified well by the authors in their response (note that the reviewer did not acknowledge the response). For these reasons, I recommend acceptance.